# The Influence of a Specific Cognitive-Motor Training Protocol on Planning Abilities and Visual Search in Young Soccer Players

**DOI:** 10.3390/brainsci12121624

**Published:** 2022-11-26

**Authors:** Andrea Casella, Emanuele Ventura, Francesco Di Russo

**Affiliations:** 1Department of Movement, Human and Health Sciences, University of Rome “Foro Italico”, 00135 Rome, Italy; 2Soccer School, Sport Club “Polisportiva Giovanni Castello”, 00145 Rome, Italy; 3Santa Lucia Foundation IRCCS, 00179 Rome, Italy

**Keywords:** cognitive-motor training, planning, visual search, soccer

## Abstract

The benefits of sport activity on cognition and especially on executive function development are well-known, and in recent years, several kinds of cognitive-motor training (CMT) have been proven effective in adults and older people. Less is known about possible CMT benefits in children. This study aims to confirm the positive influence played by CMTs on specific executive functions (planning abilities and visual search) in young soccer players. Twenty-four 10-year-old athletes were recruited and randomly assigned to the experimental (Exp) and control (Con) groups of 12 players. Both groups were trained for 10 weeks, twice a week (90 min per training), following standard soccer training (technical-tactical exercises). The Exp group, during the first training session of the week, in the last 22 min, performed the experimental treatment, which consisted of a psychokinetic CMT. Both groups were examined before and after the ten-week training using the Tower of London and WISC-IV cancellation tests. Results revealed that the Exp group, following treatment, reported significantly better scores than the Con group in all the cognitive measures. We concluded that the proposed CMT is more effective than motor training alone at improving planning abilities and visual search abilities, even in children.

## 1. Introduction

The term “developmental age” refers to the age range from 3 to 18 years, or the period of life between infancy and adolescence [1]. During this period, not only do many significant changes occur at the neuropsychological level, but the child is also expected to learn ways of adapting to and coping with reality and the daily challenges that it implies. In such contexts, the scientific literature emphasizes how a functional and proper development of executive functions (EF) is crucial for the harmonious and integral development of the human body and mind. Executive function (EF) skills are a set of neurocognitive skills that support the conscious, top-down control of thought, action, and emotion; are necessary for deliberate reasoning, intentional action, emotion regulation, and complex social functioning; and allow for self-regulated learning and adaptation to changing circumstances [2,3]. They can be defined specifically in terms of higher cortical functions assigned to behavior control and planning; skills that are essential for humans to achieve goals [2]. The influence of sports activity, and in particular the game of soccer, on the development of EFs has attracted the attention of researchers [4]. In this regard, increasing scientific evidence points to the considerable influence of specific training on the development of these functions and other cognitive functions [5,6]. For example, the authors of [7] concluded that physical activity seems to have positive effects on EF, such as planning, problem-solving, cognitive flexibility, and visual search abilities. Specifically, this research aimed to focus on the game of soccer as the most popular sport in the world that requires a fast-paced, dynamic, and strategic team sport in which highly trained motor skills and physical strength are critical for the functional enactment of skilled movement patterns in the presence of high metabolic demands [8]. In a soccer match, players must constantly assess the situation, match new events with past experiences, create new possibilities, and make quick decisions to act or inhibit planned actions. All of this occurs in a time-constrained environment while under strong emotional and motivational pressures [9]. Soccer players, belonging to elite teams, efficiently and quickly extract meaningful information through flexible visual search strategies and short periods of fixation on informative visual cues [10], anticipate key events, plan the most appropriate plan of action in advance [11], and anticipate the actions of their opponents [12]. Efficient planning abilities and visual search skills thus seem to characterize those who belong to an elite soccer team. In addition, another interesting finding is that the development of a higher EF in soccer players correlates positively with the number of goals and assists [9]. Training EF, such as visual search abilities, through specific experimental paradigms could therefore improve performance in dynamic and fast-paced ball sports such as soccer [13]. These functions, therefore, would support the athlete in adapting to and coping with a complex and multifactorial sport such as soccer. The latter involves, specifically, technical skills that have an open rather than closed nature and a series of motor operations, but especially cognitive ones such as planning, anticipation, and inhibition of action [9]. Indeed, cognitive skills, which allow constant interaction with the varied and changing situations involved in the game of soccer, often assume a role on par with, if not above, physical skills. It is no coincidence that the successful performance of players often depends on their ability to focus and anticipate the movements of opponents, to react in unexpected situations, and to take initiatives appropriate to the context [9]. In this regard, cognitive-motor training (CMT) can be used to improve soccer players’ sport and cognitive performance [13], as has recently been done for basketball [14,15]. Through a close synergy between physical and cognitive drills, the main objective is to improve players’ ability to plan, anticipate, and inhibit certain motor behaviors in relation to different game situations [13,14,15]. Specifically, such activities would stimulate situation acquisition and reading skills, planning skills, increased visual search abilities, and refine the athletes’ problem-solving process. The latter, therefore, will be required to perform specific cognitive tasks of increasing difficulty, simultaneously with physical exercises [13,16,17].

In the preparation of CMT sessions, it turns out to be of fundamental importance to devise exercises that recreate, as realistically as possible, the different game situations but that also force the athlete, at the same time, to reason in order to respond correctly [13,16,17]. Through the CMT, the athlete will gain a certain amount of experience that will then be used automatically during the game [13].

So, while it is true that the scientific literature has amply demonstrated how the game of soccer can offer significant benefits for executive and cognitive development, it is equally true that there is a paucity of research analyzing the influence of specific CMT based on psychokinetics, applied to the game of soccer, on visual search abilities and planning skills [13]. In addition, since these skills can be understood as predictors of soccer success, it turns out to be of primary importance to train and stimulate them simultaneously with technique, tactics, and athletic preparation [9]. For this reason, this research aims to develop a psychokinetic CMT specifically for young soccer players in order to improve their visual search abilities and planning skills. To this aim, a randomized-control trial protocol was adopted in 10-year-old soccer athletes.

## 2. Materials and Methods

### 2.1. Participants

Twenty-four soccer players (mean age: 10.1 years; SD = 0.4) were recruited from the soccer school of the “Polisportiva Giovanni Castello”, an elite sports club in Rome, Italy. The sample size was determined with G*Power 3.1.9.2 software [18] by estimating the effect size from Cohen’s f statistics. We set a medium expected effect size f(V) for the present mixed 2 × 2 ANOVA design at 0.26 [18]; the α level was set at 0.05, and the minimum desired power (1 − β) at 80. We adopted the following inclusion criteria: absence of any neurological and psychiatric disorders, absence of injury during the experimental session, normal or corrected-to-normal vision, and being naïve about the aim of the study. Athletes were further required to be actively involved in practicing soccer and to have at least two years of formal training in soccer. Both parents of all participants gave their informed consent before taking part in this study in accordance with the Declaration of Helsinki after approval by the local ethical committee of the University of Rome “Foro Italico” (approval code: CARD-74/2020).

### 2.2. Procedure

Participants were randomly assigned to the experimental (Exp) and control groups (Con). Each group was composed of 12 players. Participants were kept in the dark about the purpose of the experiment and did not know into which group they were inserted. The groups did not differ in age, education, socioeconomic status, or expertise (all *t*-tests resulted in *t* < 1). Both groups were trained for 10 weeks, twice a week (90 min per training), following standard soccer training (technical-tactical exercises). The Exp group, during the first training session of the week, was divided into four subgroups, made up of 3 athletes. Each subgroup, in the last 22 min of the training session, performed the experimental treatment, which consisted of psychokinetic cognitive-motor training, as described below. Before and after the 10 weeks, all players completed a specific battery of cognitive tests aimed at detecting the level of executive functioning and attention capacity.

#### 2.2.1. Cognitive Battery Tests

Measurements took place under identical conditions for all participants; pre- and post-tests were made during the first hour of the training session and 2–3 days before and after the treatment.

To test planning ability, the Tower of London (TOL) test, validated in Italian version [18] was used [19]. The TOL consisted of two identical wooden boards (30 × 7 × 10 cm) and two sets of three beads (blue, green, and red). Each board consists of three vertical pegs, where the tallest peg (Peg 1) can hold three beads at most, the middle peg, (Peg 2) can hold only two beads, and the shortest peg (Peg 3) can only hold one bead. One wooden board was used by the participant, with the beads in a standard start configuration. Another wooden board was controlled by the examiner, who demonstrated 12 target configurations [19]. These configurations have increasing difficulty levels, which are identified by requiring a minimum number of moves from two to five (Figure 1). Participants were instructed to move beads from the start configuration to the target configuration with as few moves as possible without violating task rules. The possible rule violations were placing or trying to place more beads on a peg than it can physically support and removing two beads from the peg at the same time.

From this test, two scores were computed: the total score and the number of failed attempts. The total score refers to the sum of the individual scores obtained by the subject in each of the 12 tasks (0–3 points per task), ranging from 0 to 36. Specifically, this parameter measures the ability to plan, i.e., to prepare a procedure aimed at achieving a goal and to control the execution of the procedure itself until the result is obtained. The number of failed attempts corresponds to the times the participant has violated the rules, which is an index of the ability to understand and keep in mind the rules for performing the task [19]. To evaluate participants’ visual search ability, the cancellation test of the Wechsler Intelligence Scale for Children—Fourth Edition [20] was used in the Italian version [21]. In that task, the athletes were required to identify animals on a card consisting of several items (animals and objects). The task consisted of two trials with the same objective, with a time limit of 45 s. The two trials differed only in the arrangement of the animals on the cards. In the first trial, the items were randomly arranged; in the second trial, the items were aligned in lines and columns as shown in Figure 2. The total score (the total number of identified animals in the two trials) allows us to estimate selective attention and vigilance ability.

#### 2.2.2. Intervention

The experimental intervention was defined by the training proposed by [21,22]. The Exp group executed 10 motor-cognitive tasks, one per week. Four of these tasks were administered twice; the other two were administered once. All tasks were aimed at stimulating executive functions, attention, and problem-solving using physical-cognitive exercises. The first task is depicted in Figure 3a, where three areas (A, B, and C) are divided by the dotted lines on the playing field. Three athletes were positioned in each area. The athlete in position “A” was assigned the number “1”, the athlete in position “B” assigned the number “2”, and the athlete in position “C” assigned the number “3”. In front of each area, a colored cone (red, blue, or yellow) was positioned at a distance of 1 meter inside a small training soccer net (100 × 50 × 80 cm). The task started when the coach/sports psychologist, by voice command, indicated a sequence of colors (e.g., blue, red, yellow). Initially, in positions “A”, “B”, and “C”, the athletes were asked to touch the colored cones with their hands as quickly as possible in the order provided by the voice command. Once the colored cones in positions “A”, “B”, and “C” were touched, the young athletes moved to positions “A1”, “B1”, and “C1”. Next, the athletes were asked to repeat the same task by trying to remember the color sequence initially indicated by the coach/sports psychologist. At this point, the coach/sports psychologist, by voice command, indicated a number (1–3) and a color (red, blue, or yellow). The athlete corresponding to the number indicated by voice feedback assumed the role of defender, and the other two athletes assumed the role of attackers. The attackers had to score a goal in the small soccer net, corresponding to the last color indicated by the coach/sports psychologist. In order to increase the difficulty of the proposed exercise, some variations were included. In positions “A1”, “B1”, and “C1” the athletes were asked to touch the cones in the reverse order of the sequence of colors indicated in positions “A”, “B”, and “C”. Each colored cone was, subsequently, associated with a number. In this case, the voice feedback did not indicate a directly observable sequence of colors but a sequence of numbers that the young athletes had to memorize and reproduce. The second task is depicted in Figure 3b, where three horizontal segments of distinct colors were made on the playing field. So, the first athlete was placed in the black segment. The second was placed in the red segment. The third was placed in the blue segment. Before the athletes began training, they were instructed that the black segment had been associated with the number 3, the blue segment with the number 1, and the red segment with the number 2. The athlete positioned in the black segment had the task of performing ball-handling drills alternated with technical skills as desired. The athlete positioned in the blue segment had the task of imitating the behavior enacted by the athlete positioned in the black segment. The athlete positioned in the red segment represented an interfering stimulus. The latter had to perform a ball-conducting exercise between the red cones. The three athletes were required to perform the task until the coach/sports psychologist, by voice command, indicated a number (1–3). The athlete who was in the position corresponding to the number indicated by the voice feedback assumed the role of defender, and the other two athletes assumed the role of attackers. The latter had to score a goal in one of two small soccer nets located at the goal area’s lower vertices at 4.5 m. The third task is reproduced in Figure 3c. Three colored cones were placed on the playing field. Three smaller soccer goals (100 × 50 × 80 cm) were spaced approximately 5 meters apart. Each soccer goal was associated with a number from one to three. The coach/sports psychologist, by voice command, indicated a sequence of colors to touch as quickly as possible and a number corresponding to the soccer goal in which to make the goal.

The fourth task is reproduced in Figure 4a, consisting of a small playing field created within the penalty area of the 11-a-side soccer field. The 12 young athletes were divided into two teams. The first team consisted of “reds” and “yellows”, and the second team consisted of “blues” and “greens.” Both teams were asked to score as many goals as possible in the 22-minute training session. However, each team was required to make at least three passes before they could try to score a goal. In addition, passing had to be done only between teammates that have different colors (for example, each “red” athlete could only pass the ball to a “yellow” athlete, and vice versa; passing between teammates of the same color was not possible). The fifth task is reproduced in Figure 4b. Three colored cones were placed on the playing field. At a distance of approximately 1.5 m apart, three more cones of the same colors were placed. Two smaller soccer goals (100 × 50 × 80 cm) were placed approximately 4 m apart. Each soccer goal was associated with a name: “right” and “left.” The coach/sports psychologist, by voice command, indicated a sequence of colors to be touched as quickly as possible in the first row of cones. The young athlete should next touch the cones in reverse order, in the second row of colored cones, memorizing the order. He will finally have to score a goal in the soccer goal indicated by the voice command. The sixth task is reproduced in Figure 4c. Small red targets and some colored cones were placed on the playing field. The red-colored targets formed four distinct numbers from one to four. Two different colored cones were placed in front of each number at a distance of about 1.5 m apart. Two smaller soccer goals (100 × 50 × 80 cm) were placed about 5 m apart. The soccer goals were associated with two names: “right” and “left.” The coach/sports psychologist, by voice command, indicated a number, a sequence of colors to be touched as quickly as possible, and in which goal to score the goal (e.g., “one—red—blue—red—left”).

The performance during the intervention could not be measured because of a lack of effective instrumentation. We ensured that all participants in the Exp group performed the intervention by dividing the Exp group into three sub-groups and directing task execution monitoring.

### 2.3. Data Analysis

The TOL (total score and failed attempts) and WISC-IV cancellation (total score) data were analyzed using a mixed 2 × 2 analysis of variance (ANOVA) with an independent group factor at two levels (Exp vs. Con group) and a repeated treatment factor (pre- vs. post-test). The partial eta squared (η_p_^2^) was also reported as a measure of effect size. Post-hoc comparisons were performed using the Bonferroni correction, and the α threshold was set to 0.05. Normal distribution and equal variance were checked through the Shapiro–Wilks and Levene’s tests. All statistical analyses were performed using the Statistica 12.0 software (StatSoft, Inc., Tulsa, OK, USA).

## 3. Results

Statistical data are reported in Table 1. The ANOVA on the TOL total score revealed significant effects of group and treatment, and the interaction between these two factors was also significant (Figure 5). Post-hoc comparisons revealed that, while in the pre-test the groups made similar scores, in the post-test the Exp group score was higher (*p* < 0.001) than the pre-test and higher than the score of the Con group (*p* < 0.001) in both the pre- and post-tests, which did not differ from each other.

Comparable results were obtained for the “failed attempts” outcome of the TOL. Again, both groups showed similar values on the pre-test, but in the post-test, the Exp group displayed fewer errors than the Con group in both the pre- (*p* < 0.001) and post-test (*p* < 0.001), which did not differ from each other (Table 2 and Figure 6).

Finally, in the ANOVA of the WISC-IV cancellation, the group factor was not significant, while the treatment factor and the interaction were significant. A post-hoc comparison of the interaction (Table 3 and Figure 7) showed that both groups had similar values at the pre-test. However, only the Exp group improved at the post-test (*p* < 0.001), displaying higher scores than the Con group (*p* <0.025).

## 4. Discussion

Results in the TOL test indicated that, while the control group did not show any change after the standard training, the experimental group improved its performance in both planning ability and rule compliance. Regarding the WISC-IV cancellation test, both groups improved their visual search skills, but the experimental group’s improvement was larger, resulting in a higher score than the control group in the post-test. These results confirm the literature, showing that specific cognitive-motor workouts positively influence the targeted cognitive functions [4,13,16,17,23]. The improvements obtained by the experimental group may be interpreted in terms of an increase in the abilities to plan to prepare a procedure aimed at achieving a goal and to control the execution of the procedure (detected by the variable total score in the TOL) and in the ability to understand and keep active in memory instructions useful for achieving goals (detected by the variable fail attempts in the TOL). In the same vein, [4,24] found that specific sports activities related to the game of soccer enhance the planning and inhibition skills required by the Tower of London task. This result is consistent with characterizations of soccer as a “thinking game” involving open skills. Indeed, kicking a moving ball is a gesture that engages high-level cognitive processes and requires perspective control, perceptual skills, and motor coordination [25]. In fact, in a soccer game, players are required to constantly assess the situation they are in to match new events with past experiences, create new possibilities, and make quick decisions about whether to act or inhibit planned actions. All of this occurs in a time-limited environment while being subjected to strong emotional and motivational pressures. It is no accident that soccer players show significantly higher levels of executive functioning than sedentary controls [9].

Regarding the results obtained from the WISC-IV cancellation test, a significant improvement in both groups’ visual search skills could be found. However, the effect of the experimental treatment was stronger, allowing a higher score than the control group. This finding is corroborated by long-standing research, which showed that soccer players belonging to professional teams, efficiently and quickly grasped meaningful information through flexible visual search strategies and short periods of fixation on informative visual cues [26]. Soccer training can help players focus on key events and the most appropriate action for the situation [11], allowing them to functionally anticipate their opponents’ actions [12]. So, it is clear how the game of soccer can provide significant benefits for the cognitive development of young athletes. However, the combination of standard soccer training with the proposed psychokinetic treatment may stimulate attentional function even more than physical training alone. In light of the present and previous experimental evidence offered [13], the introduction of specific cognitive-motor training in soccer (such as in other sports [16,17] not only improves sports performance, but such combined training could also be used as valuable support for mental training to prevent burnouts and injuries due to training sessions of excessive intensity and duration [27]. In fact, soccer players’ performance and injury prevention are typically analyzed from physical-tactical, biomechanical, and metabolic perspectives [9]. However, as extensively described in the literature, executive functions and visuospatial skills have been shown to be critical for efficiency and well-being in sports [14,15,28].

However, as with basketball [16,17], there has been a need to further investigate the effects of specific cognitive-motor training on soccer performance. This seems particularly relevant as it has been suggested that practical training approaches should consider ecological validity, transferability, and action-perception coupling [29,30,31,32]. Along the same theoretical lines, [33] investigated the effects of cognitive training on sports performance. A careful analysis of the results shows substantial agreement with previously presented experimental evidence [12,16,17,34,35,36]. Indeed, even in that case, it was possible to show a significant improvement in cognitive domains, particularly in response time to stimuli in the peripheral field, in subjects undergoing the experimental treatment (cognitive-motor training) compared to controls.

## 5. Conclusions

Considering the mutual influence between sports practice and executive function training [4,9,13] and that 10-year-old children undergo a natural biological maturation of the executive functions [37,38,39], we believe that including CMT at this age could be important because both sports practice and training for executive abilities may contribute to executive function maturation and improve sports performance.

Soccer-specific cognitive-motor training could become a valuable support for soccer schools because it may especially increase specific executive and attentional functions, which are fundamental in soccer and everyday life.

## 6. Limits

The presented results have some limitations: (a) the participants were only male, limiting the generalization; (b) the sample size was the minimum to allow a medium effect size; (c) sport performance measures were not taken; and (d) the absence of a control group made up of subjects who did not participate in sporting activities would make it possible to ascertain that the progress shown by those who took part in the standard training sessions actually depended on their practice of sport and not on other factors. In the future, it would be advisable to replicate this research with more participants, an additional control group of non-athletes, and both motor and cognitive testing in order to consolidate its validity, expand the possibility of generalizing the results, and involve female soccer teams.

## Figures and Tables

**Figure 1 brainsci-12-01624-f001:**
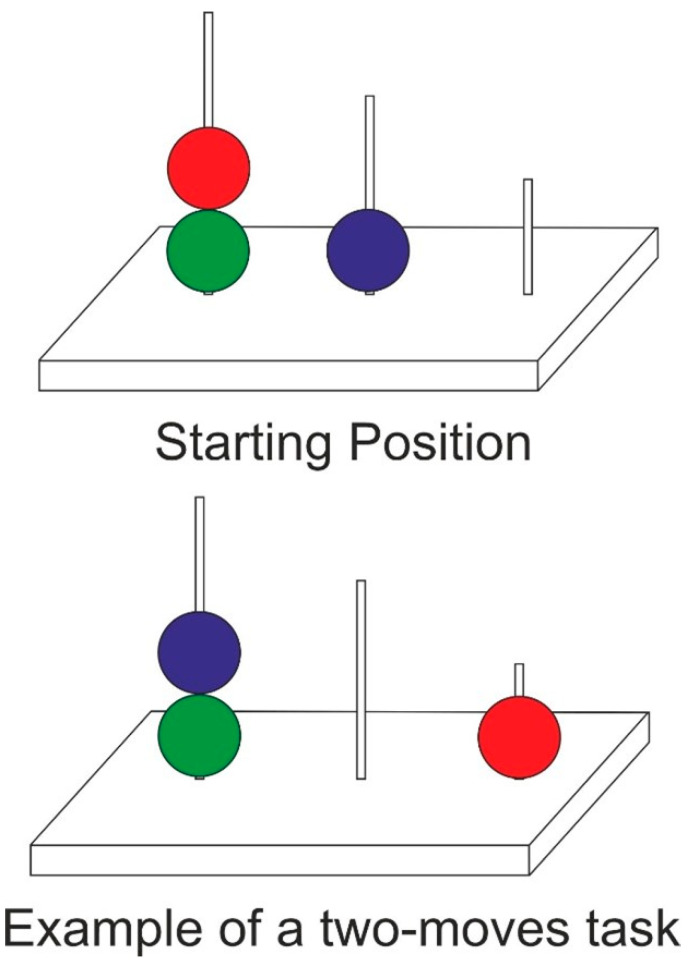
An example of a TOL task with a starting configuration and a simple target configuration to be reached with two moves.

**Figure 2 brainsci-12-01624-f002:**
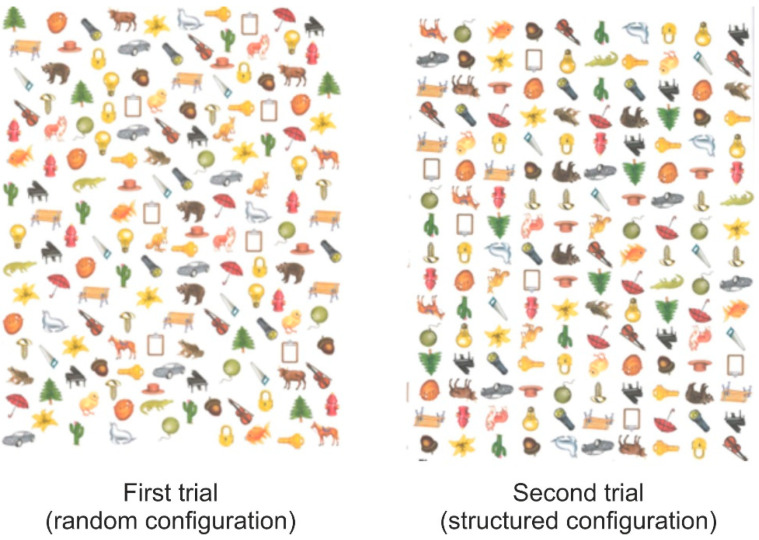
Cancelation test of the WISC-IV scale.

**Figure 3 brainsci-12-01624-f003:**
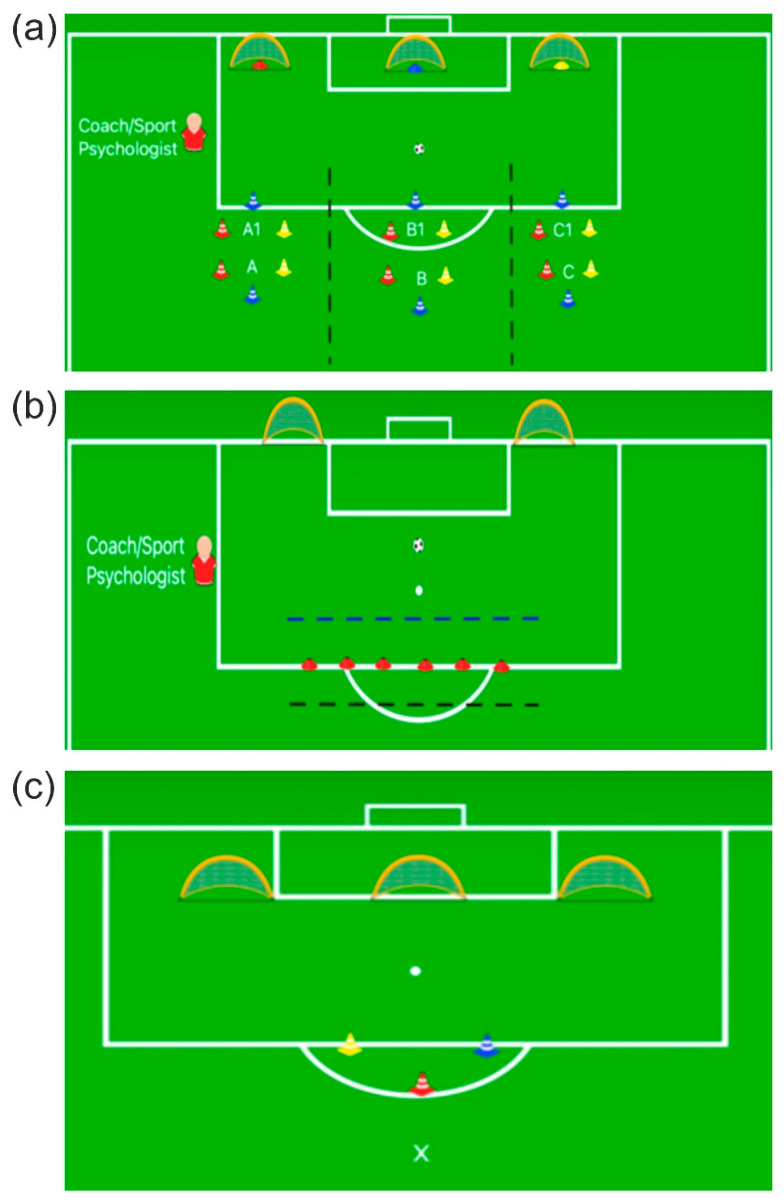
Representation of the first (**a**), the second (**b**), and the third task (**c**) of the experimental treatment, which were repeated twice).

**Figure 4 brainsci-12-01624-f004:**
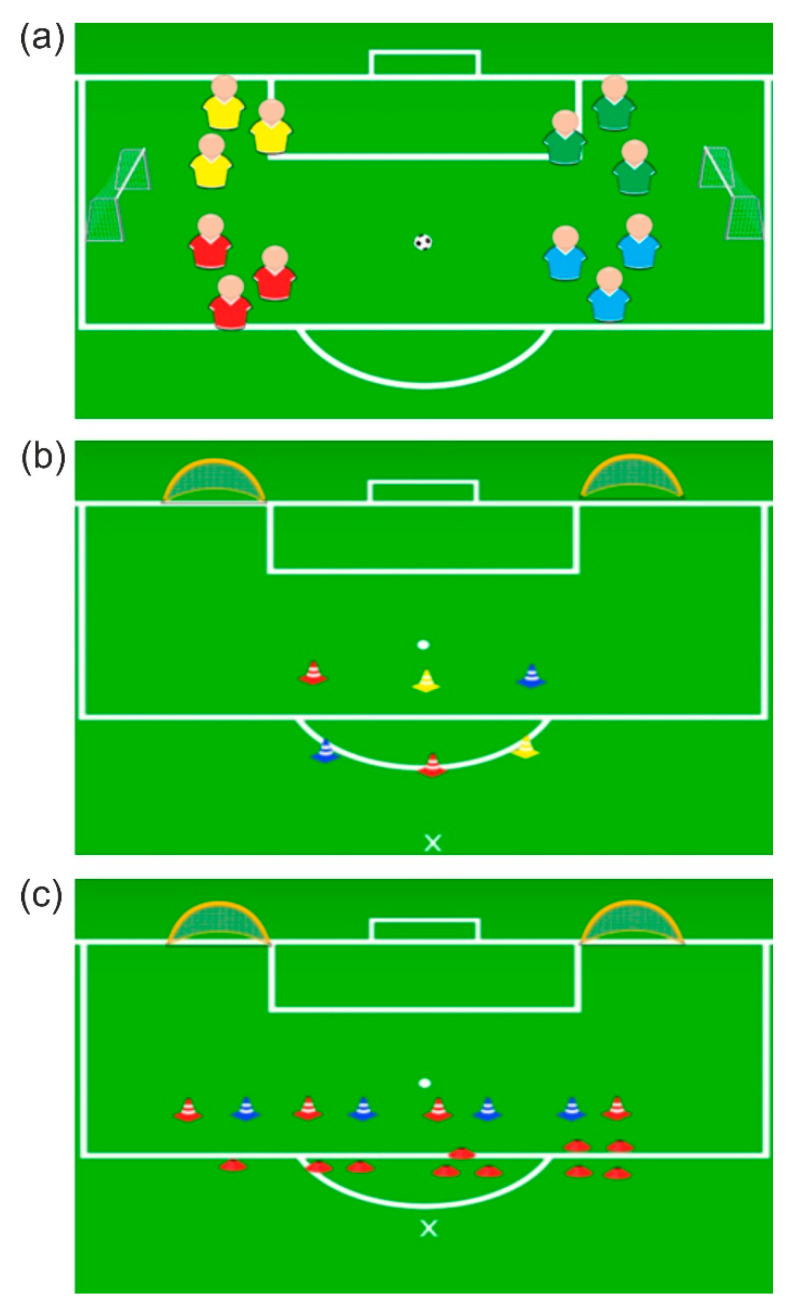
The other three tasks of the experimental treatment. The first (**a**) was performed twice, while the second (**b**) and third (**c**) were performed only once.

**Figure 5 brainsci-12-01624-f005:**
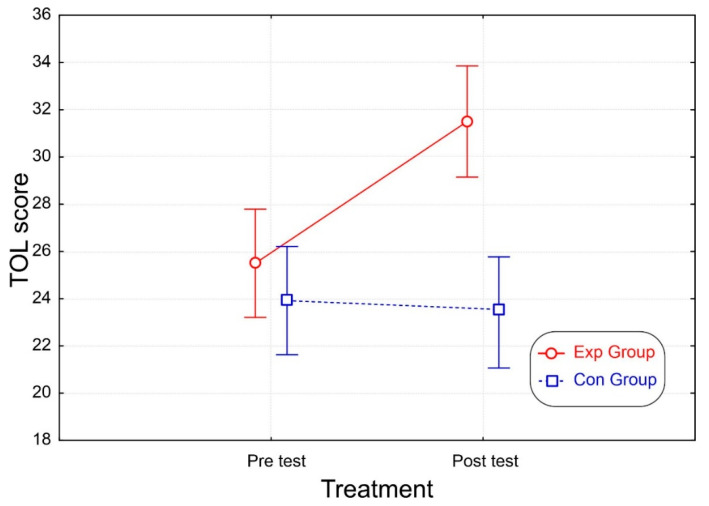
Total score in the TOL test for the two groups before and after the treatment. The vertical bars denote the 95% confidence interval.

**Figure 6 brainsci-12-01624-f006:**
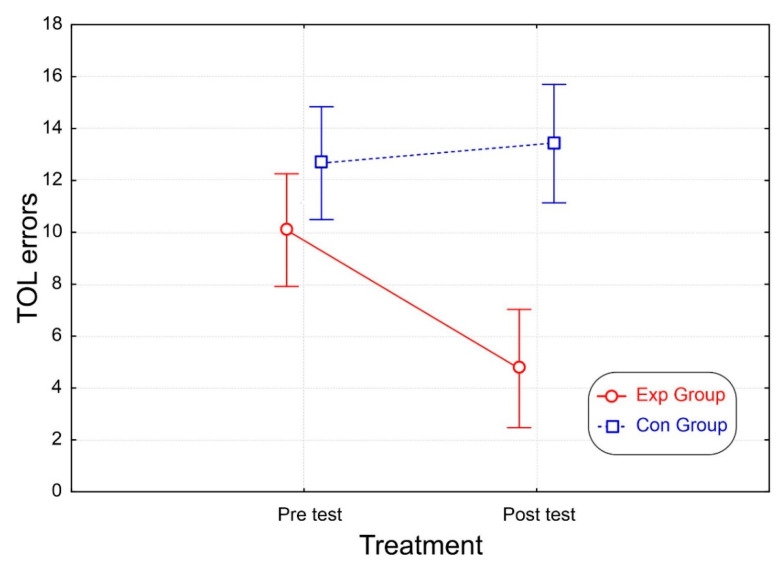
Results of failed TOL test attempts. The vertical bars denote the 95% confidence interval.

**Figure 7 brainsci-12-01624-f007:**
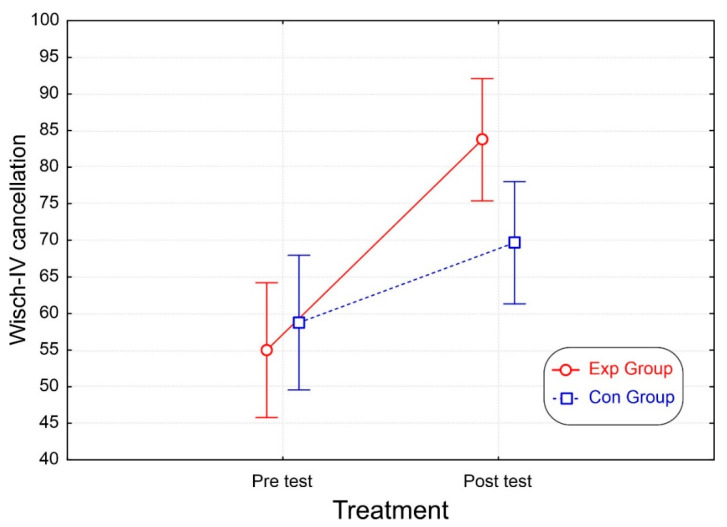
The number of correct items detected in the WISC-IV cancellation test. The vertical bars denote the 95% confidence interval.

**Table 1 brainsci-12-01624-t001:** ANOVA data for the TOL total score. Degrees of freedom (DOF).

TOL	DOF	F	*p*	η_p_^2^
Group	1, 22	13.6	0.001	0.381
Treatment	1, 22	9.6	0.005	0.305
Group × Treatment	1, 22	13.4	0.001	0.379

**Table 2 brainsci-12-01624-t002:** ANOVA data for the TOL failed attempts. Degrees of freedom (DOF).

TOL	DOF	F	*p*	η_p_^2^
Group	1, 22	18.6	<0.001	0.456
Treatment	1, 22	8.6	0.007	0.282
Group × Treatment	1, 22	15.2	<0.001	0.409

**Table 3 brainsci-12-01624-t003:** ANOVA data of the WISC-IV cancellation. Degrees of freedom (DOF).

TOL	DOF	F	*p*	η_p_^2^
Group	1, 22	1.0	0.332	0.042
Treatment	1, 22	45.1	<0.001	0.672
Group × Treatment	1, 22	9.1	0.006	0.293

## Data Availability

Data are available from the corresponding author upon request.

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
