# Peer review of "The Influence of a Specific Cognitive-Motor Training Protocol on Planning Abilities and Visual Search in Young Soccer Players"

_brainsci, 2022, doi:10.3390/brainsci12121624_

Round 1

Reviewer 1 Report

Training to influence cognitive functioning has been discussed many times regarding various groups, including athletes. I must admit that the results obtained by the authors raise my concerns and suspicions; even the effects of much longer and more complex interventions are usually much smaller than those the authors present. Sometimes a near transfer (improvement in an untrained task) and, very rarely, a far transfer (improvement in a different function altogether) can be observed. Meanwhile, the authors show spectacular effects after only 5 training sessions. So I have serious doubts about the methodology. 

- precisely how groups were selected

- whether the measurements took place under identical conditions for all subjects

- whether the subjects knew which groups they were in

- how the authors explain their spectacular results, which are very different from many previously obtained results

-why there is no practiceeffect in the control group (performing the same task for the second time is easier than the first time, so post-test scores should be slightly higher than in the pretest)

- how was the .6 effect size determined? According to the authors, this is a medium effect, but this is incorrect; 0.6 is a large effect (e.g. Cohen (1988) Statistical Power Analysis for the Social Sciences).

- I believe that the groups are too small and not diverse enough to determine the effectiveness of the intervention

Moreover, there are inconsistencies in the descriptions:

in the abstract - both groups were trained for 10 weeks

in the text - both groups were examined before and after a five-week training

Author Response

Response. Maybe the effects seem spectacular and raised doubts in the Reviewer because we committed a typo in the abstract about the training duration. The soccer training sessions were actually twenty including ten experimental sessions, not five.

- precisely how groups were selected

Response: As specified in paragraph 2.2 Procedure, the participants were randomly assigned to the experimental and control group.

- whether the measurements took place under identical conditions for all subjects

Response: As suggested we now specified in paragraph 2.2.1 Cognitive battery tests, that the measurements took place under identical conditions for all participants, pre- and post-tests were made during the first hour of the training session 2-3 days before and after the treatment.

- whether the subjects knew which groups they were in

Response: As suggested we now specified in paragraph 2.2 Procedure, that participants were kept in the dark about the purpose of the experiment and did not know into which group were inserted.

- how the authors explain their spectacular results, which are very different from many previously obtained results

Response. As anticipated above, the results are not so “spectacular” if we consider that the training sessions were actually ten including ten experimental sessions and not just five. As stated in the Discussion section, the results confirmed all those studies showing a positive influence of cognitive motor-training protocols on athletes' executive functions. Specifically, Alesi et al. 2016 reported a significant increase in the TOL score in young football players involved in a cognitive-motor exercise program. Lucia et al., 2021 and 2022, found benefits of a cognitive-motor training on anticipatory and executive functions of basket players.

-why there is no practice effect in the control group (performing the same task for the second time is easier than the first time, so post-test scores should be slightly higher than in the pretest)

Response. Even if non-significant, the practice effect was actually present for the visual search ability in the cancelation test. The lack of practice effects in the control group was only present for the TOL test likely because 10 weeks between tests are not enough to allow learning for this kind of procedural task.

- how was the .6 effect size determined? According to the authors, this is a medium effect, but this is incorrect; 0.6 is a large effect (e.g. Cohen (1988) Statistical Power Analysis for the Social Sciences).

Response. We thank the Reviewer for noticing this typo that has been fixed, we meant 0.26, which is a medium effect size typically used for mixed ANOVA designs (Faul et al., 2009).

- I believe that the groups are too small and not diverse enough to determine the effectiveness of the intervention

Response. The sample size has already been included within the limits of this study. However, the results obtained can be used as a starting point to further investigate this topic.

Moreover, there are inconsistencies in the descriptions: in the abstract - both groups were trained for 10 weeks in the text - both groups were examined before and after a five-week training

Response. We fixed the typo; the training duration was 10 weeks.

Reviewer 2 Report

Thank you for the opportunity to review this article. It is of interest from the perspective of cognitive and developmental psychology as well as sport psychology. The strength of the article is, in my opinion, the thorough description of the procedure and the research tools used. A very worthwhile contribution are the tasks used in psychokinetic cognitive-motor training. These too are well described and illustrated. The authors also aptly note that a shortcoming of the research conducted is the lack of knowledge of how such training results not only in cognitive test performance, but also in the soccer player's game efficiency.

My doubts are raised by the conclusion that "the proposed CMT is more effective than motor training alone to improve planning abilities and visual search abilities even in children". In fact, we do not know (based on the research conducted) whether the standard motor training of soccer players actually increases these cognitive abilities to any extent. With regard to TOL, the control group showed no improvement. In terms of Wisch-4, on the other hand, the control group improved their performance, although to a lesser extent than the experimental group. However, we cannot be sure here that the observed improvement in the control group is the result of motor training and not of repeated measurement (learning effect). Indeed, there was no control group with no training and only two measurements with the same cognitive tool. Thus, while we can infer the impact of psychokinetic training on cognitive functioning from the results of the studies carried out in the adopted model, we cannot evaluate the impact of standard soccer training alone. I believe that the conclusions need to be modified and that the issue described above should be addressed in the discussion and/or limitations of the study. 

Author Response

We thank the Reviewer for the appreciation.

Response. We agree with the Reviewer’s concern, and accordingly, we now modified the discussion section and added as a limitation the lack of an additional control group not practicing sport. However, this point will be an element for future research.

Reviewer 3 Report

It's a very curious study and the manuscript is well written (I enjoyed reading it!). I just have few general questions:

1. I think is very interesting this dual influence from sport to executive functions and from executive functions to sports. It is briefly argued at the end of the introduction but I think it should be extended in the discussion too. For example, athletes might be prepared to an increased improvement of their executive functions because of the mutual support between the two.

2. How the intervention was defined? and how about the timing/repetitions of the intervention administration? Did you run a pilot? Was the performance during the interventions measured? How can you ensure that subjects in the exp group all performed the intervention? I think this should be better defined in the text.

3. I understand the goals of the study, but why the authors chose not to have a pure control group of no-athletic kids?

4. How about adults? in the Introduction, the authors refer to previous studies with adult athletes, but not with the same intervention. Was the intervention specifically developed for kids (see my question 2 for the intervention in general)? Would you expect similar results in adults?

5. " further investigations will be needed to functionally investigate" it's a repetition

6. I suppose it's because of limit in words number, but I would add all the tasks of the intervention in the main text (not supplementary). Or, alternatively, might be better to summarize them all in the main text, and then define them all in details in supplementary. 

Author Response

We thank the Reviewer for the appreciation.  
  1. I think is very interesting this dual influence from sport to executive functions and from executive functions to sports. It is briefly argued at the end of the introduction but I think it should be extended in the discussion too. For example, athletes might be prepared to an increased improvement of their executive functions because of the mutual support between the two.

Response. As suggested, we added in the discussion section a paragraph about the dual influence between sport practice and executive functions training.

  1. How the intervention was defined? and how about the timing/repetitions of the intervention administration? Did you run a pilot? Was the performance during the interventions measured? How can you ensure that subjects in the exp group all performed the intervention? I think this should be better defined in the text.

Response. As requested, we now stated in the 2.2.2 Intervention section that the experimental intervention was defined in accordance with the training proposed by Pietrocini & Rubba (2014) and Calligaris (2013). We also stated that performance was not measured because of the lack of effective instrumentation and that we ensure that participants in the exp group all performed the intervention, dividing the exp group into sub-groups of three and directing monitoring of the task execution.

  1. I understand the goals of the study, but why the authors chose not to have a pure control group of no-athletic kids?

Response. The lack of an additional control group of no-athletic kids has been considered in the study limitations.

  1. How about adults? in the Introduction, the authors refer to previous studies with adult athletes, but not with the same intervention. Was the intervention specifically developed for kids (see my question 2 for the intervention in general)? Would you expect similar results in adults?

Response. The present intervention was specifically developed for children, but we were also inspired also by the numerous adult studies on CMT. However, following an ample literature search, it emerged that only a small number of studies dealt with children. In agreement with Lucia et  al. (2021, 2022), we could expect comparable results in the adult population, but the CMT should be calibrated, increasing the task difficulty and speed, and also using interactive devices such as those used by Lucia et  al. (2021, 2022)

  1. " further investigations will be needed to functionally investigate" it's a repetition

Response. We removed that statement.

  1. I suppose it's because of limit in words number, but I would add all the tasks of the intervention in the main text (not supplementary). Or, alternatively, might be better to summarize them all in the main text, and then define them all in details in supplementary. 

Response. As suggested, we now added all the tasks of the intervention in the main text.